# Advancement of Mechanical Properties of Nickel-Titanium Rotary Endodontic Instruments by Spring Machining on the File Shaft

**DOI:** 10.3390/ma13225246

**Published:** 2020-11-20

**Authors:** Sangmi Ahn, Jung-Hong Ha, Sang Won Kwak, Hyeon-Cheol Kim

**Affiliations:** 1Department of Conservative Dentistry, School of Dentistry, Dental Research Institute, Dental and Life Science Institute, Pusan National University, Yangsan 50612, Korea; smahn4009@gmail.com; 2Department of Conservative Dentistry, School of Dentistry, Kyungpook National University, Daegu 41940, Korea; endoking@knu.ac.kr

**Keywords:** fatigue, fracture, resistance, spring, structure, torsion fracture

## Abstract

Nickel-titanium (NiTi) endodontic rotary instruments are used extensively in root canal procedures by both general dentists and specialists. However, their vulnerability to fracture is the major reason for clinicians’ concern regarding their use. The objective of this study was to investigate the potential effects of spring machining of the file shaft on the fatigue and torsional resistances of NiTi rotary instruments. Three types of NiTi rotary systems with (S) and without (NS) spring machining were used in this study (*n* = 15 each): a spring file (SPR; #25/.06, SPR-S, SPR-NS), a ProTaper Next X2 (PTN; #25/variable taper, PTN-S, PTN-NS), and a ProTaper Gold F2 (PTG; #25/variable taper, PTG-S, PTG-NS). Spring machining was adjusted on the 6 mm of each file system’s shaft via a laser cutting process. The number of rotation cycles until fracture (i.e., cyclic fatigue resistance), ultimate torsional strength, the distortion angle, and the toughness of each subgroup were estimated with specially designed devices. The results were analyzed using a paired *t*-test at a significance level of 95%. NiTi rotary instruments with spring machining exhibited a higher cyclic fatigue resistance than instruments without spring machining. The groups with spring machining exhibited a higher toughness and larger distortion angle than the groups without it (*p* < 0.05). In conclusion, spring machining on the shank of NiTi instruments may provide a stress-bearing area and attenuate the torsional and cyclic fatigue of NiTi rotary instruments.

## 1. Introduction

Nickel-titanium (NiTi) endodontic rotary instruments are used extensively in root canals by both general dentists and specialists. Using NiTi rotary instruments, the canal aberration by instrumentation can be reduced owing to their flexibility, and the procedural time can also be reduced [1]. However, these instruments’ vulnerability to separation is the major reason for clinicians’ concern regarding their use [2]. The two major mechanisms that lead to the separation of instruments are cyclic and torsional fatigues [3]. Cyclic fatigue is caused by repetitive compressive and tensile stresses acting on a file rotating in a curved canal, and torsional failure occurs when the tip of the instrument binds but the shank of the file continues to rotate [3,4,5].

Thus, various efforts, such as geometric changes, surface and/or heat treatments, and changing protocol, have been directed toward enhancing the cyclic fatigue resistance and torsional resistance of NiTi rotary instruments [6,7,8,9,10,11,12,13,14,15,16,17]. Various heat-treated NiTi rotary instruments were recently developed and introduced. Their superior cyclic fatigue resistance compared with the conventional NiTi alloy was proven in [6,7]. The controlled memory wire, which exhibits a high austenitic transformation finishing temperature (Af), was introduced and exhibited an enhanced cyclic fatigue resistance and higher flexibility compared to existing NiTi rotary instruments [8,9,10,11]. In addition to the alloy, modifications to the geometry (e.g., pitch and cross section) have been attempted, improving the fracture resistance [12]. Isik et al. reported that the shaft length itself may affect the fracture resistance [13]. According to previous studies, the torsional deformation and/or fracture of NiTi rotary instruments can be reduced by reducing the pitch (increasing the number of threads) and increasing the cross-sectional area [14,15].

Recently, during a trial to enhance the cyclic and torsional fatigue resistance of NiTi rotary instruments, a spring file system (DenFlex, Seoul, Korea) was developed. A unique spring structure is featured in the spring file and the shaft of the NiTi rotary instruments via laser cutting (Figure 1). The manufacturer claims that the spring structure may act as a buffer and reduce stress generation during instrumentation. Furthermore, this technique has a wide range of applicability, which can be applied to any type of NiTi rotary instrument.

However, there are no scientific data regarding the mechanical properties of spring-machined NiTi rotary instruments, and spring machining may influence the fracture resistance of the instruments. Thus, the objective of this study was to evaluate the potential effects of spring machining on the cyclic fatigue and torsional resistance of various NiTi rotary instruments. The null hypothesis was that there would be no difference in cyclic fatigue and torsional resistance in NiTi file systems according to the presence of spring machining.

## 2. Materials and Methods

Three types of NiTi rotary systems with (S) and without (NS) spring machining were used in this study (*n* = 15 each): a spring file (SPR; #25/.06) with and without spring machining (SPR-S, SPR-NS) (Figure 1), a ProTaper Next X2 (PTN; #25/variable taper, Dentsply Sirona, Ballaigues, Switzerland) with and without spring machining (PTN-S, PTN-NS), and a ProTaper Gold F2 (PTG; #25/variable taper, Dentsply Sirona) with and without spring machining (PTG-S, PTG-NS). The two file systems (PTN and PTG) were included to evaluate whether the spring structure actually works for other commercially introduced file systems. The PTN system is a representative file system with an eccentric center of rotation due to the off-center design of its cross section, while PTG is a conventional rotary instrument with a convex triangular cross section [18,19]. SPR and PTG are made of heat-treated NiTi controlled-memory wire, while PTN is made of m-wire NiTi alloy. The length of the file was 25 mm in all cases.

Spring machining was adjusted on the file shaft via a laser cutting process using a custom manufacturing device from the company DenFlex with the laser being generated by IPG Photonics (Burbach, Germany). The spring structure was adapted to contain a 6-mm-long spring in the shaft with a 2600° spring coil (Figure 1).

The torsional resistance was measured using a custom-made device (AEndoS; DMJ system, Busan, Korea) according to the American National Standard/American Dental Association specification no. 28 [20] and ISO specification 3630-1 [21] with some modifications. The file tip of the apical 3-mm part was rigidly fixed between two resin blocks made of polycarbonate (Figure 2A). Each file was driven clockwise at 2 rpm speed using a computerized program designed for the device until a file fracture occurred. The ultimate strength (Ncm) and distortion angle until fracture (°) were recorded at a rate of 50 Hz, and the toughness (°·Ncm) was calculated using the area under the plot of the distortion angle (*X* axis) and the torsional load (*Y* axis) using Origin v6.0 Professional (Microcal Software Inc., Northampton, MA, USA).

The cyclic fatigue resistance was evaluated using another custom-made device (EndoC; DMJ system) with a simulated canal block (Figure 2B). Using the device, repeated simulations in a curved canal with a 35° angle of curvature were performed. The instruments from each subgroup (*n* = 15 each) were rotated at 300 rpm in the dynamic mode, with a 4-mm-deep up-and-down pecking motion performed by a computerized program designed for the device. At the moment when the instrument fracture occurred, the time for fracture was recorded using a chronometer. The number of cycles to failure (NCF) for each instrument was calculated by multiplying the total time (s) to fracture by the rotation rate (5 revolutions per second, 300 rpm). The length of the fractured file tip was measured using a digital micro-caliper (Mitutoyo, Kawasaki, Japan).

After the cyclic fatigue and torsional resistance tests, five fractured fragments were randomly selected for observation and evaluation of the topographic features of the fractured surfaces using field-emission scanning electron microscopy (FE-SEM; JSM-7200F; JEOL Ltd., Tokyo, Japan).

Mean and standard deviations were calculated for each group, and a paired *t*-test was used to compare torsional and cyclic fatigue resistance within a file system with or without spring machining. The statistical analysis was performed by SPSS v25.0 (IBM Corp, Armonk, NY, USA) and the significance level was set at *p* < 0.05.

## 3. Results

The mean and standard deviations of the cyclic fatigue resistance and torsional resistance for each instrument group are presented in Table 1 and Table 2.

The NCF values were significantly higher for the groups with spring machining than for the groups without spring machining in all file systems (*p* < 0.05). The NCF value was highest for PTG, followed by PTN and SPR. Additionally, the group with spring machining exhibited a higher toughness and larger distortion angle than the group without spring machining in all file systems, and the difference was statistically significant (*p* < 0.05). The toughness was highest for PTG, followed by SPR and PTN (*p* < 0.05). The distortion angle was largest for SPR, followed by PTG and PTN (*p* < 0.05). However, there was no statistically significant difference between the spring-machined group and the control with regard to the ultimate strength (*p* > 0.05).

The SEM topographic examination revealed the typical appearances of two failure modes: cyclic fatigue fracture (Figure 3) and torsional fracture (Figure 4). There were no specific differences between the file groups with and without spring machining. Specimens from the cyclic fatigue tests exhibited crack initiation area(s), crack propagations, and overloaded fast fracture zone(s) in the cross-sectional views, as well as microcracks along the machining groove on the longitudinal section (Figure 3). Specimens from the torsional resistance tests exhibited typical features, such as circular abrasion marks and skewed dimples near the center of rotation in the cross-sectional views, and exhibited the unwound helix of the file flute and slipped structure (microcracks) in the longitudinal aspects of the file (Figure 4).

## 4. Discussion

NiTi rotary instruments are commonly used in contemporary dentistry, owing to their various advantages, including their low technical sensitivity, short working time, and low incidence of postoperative sensitivity [1,22,23]. However, the higher risk of separation compared with hand instruments is a major concern [2,24]. Therefore, the development of NiTi rotary instruments aims to reduce the fracture risk, and the performance has been improved from various treatments and studies. For this purpose, spring machining of the shaft was attempted.

To evaluate the effect of spring-machined NiTi rotary instruments on the prevention of instrument separation, several values related to cyclic fatigue and torsional resistance were examined for commercial instruments with spring machining. Then, the mechanical properties of NiTi rotary instruments with and without spring machining were compared. The instrument size of #25 was used throughout the present study because it is most commonly used size for the canal preparation procedure.

According to the results, the NiTi rotary instruments with spring machining exhibited a higher cyclic fatigue resistance, greater toughness, and a larger distortion angle than those without spring machining. Although the effect differed according to the evaluated parameters and alloys, it appeared that spring machining facilitated the distribution of stress, which was absorbed by the spring structure on the shank. Therefore, the null hypothesis was rejected.

The NCF was higher for the spring-machined groups than for the control groups, which appeared to be caused by the absorption of stress by the spring-machined part of these instruments. Regarding the types of NiTi rotary file, the NCF was largest for PTG with gold wire, followed by PTN (M-wire). This result agrees with previous studies comparing the cyclic fatigue resistance between PTG and PTN [25]. The present study focused on the effectiveness of spring machining. PTN and PTG systems were tested as another verification procedure of whether the spring structure really works for other commercially marketed file systems. The differences between the file systems are basically decided by their geometric differences and alloy [7,9,11,14,26,27,28].

During the torsional resistance test, the spring-machining part of each instrument was distorted just before the instrument broke, and it appeared that the spring-machined site could absorb the torsional stress when the load was applied. Indeed, it was predicted that spring machining could enhance the torsional resistance of NiTi rotary instruments. As expected, the distortion angle was larger for the spring-machined group than for the non-machined, non-spring groups, and the stress absorption by the spring-machined structure appeared to contribute to the extension of the rotational angle before the instrument fractured. However, while the clamped area was set at the D3 level in the experiment, torsional stress in clinical situations can be generated or accumulated in the whole length of the flute. Thus, the fracture of the cutting area may still happen before the fracture of the spring portion.

Meanwhile, the ultimate torsional strength was not significantly different between the instruments with and without spring machining. The ultimate torsional strength is usually determined mainly by the cross-sectional area [29,30], and is affected by the taper, design, and alloy [4,31]. As these factors were not influenced by the spring machining, the effect of the spring machining on the ultimate torsional strength appeared to be insignificant. Furthermore, as the ultimate torsional strength was not changed and the distortion angle was increased, the toughness was enhanced for the instruments with spring machining.

The improvements in the toughness and distortion angle were larger for PTG than for SPR and PTN. It appeared that the effect of spring machining depended on the alloy of the NiTi instruments. Further studies are needed to confirm this.

In addition to the enhancement of the mechanical properties of the NiTi instruments, spring machining may have other merits. It was reported that a longer shaft may have higher torsional resistance than a shorter one [13]. However, a longer-shafted file is inherently difficult to use in the molar area. The clinical application of NiTi rotary instruments to the posterior area is occasionally difficult, particularly for patients who have narrow occlusal clearance or limitations in opening their mouths. In these cases, improper application of NiTi rotary instruments can result in instrument separation due to excessive bending and/or an erroneous stress concentration. Thus, NiTi rotary instruments with spring machining would be helpful in these situations, as the machined part can be bent inside the mouth and can be efficiently applied to molars.

By using a spring-machined NiTi rotary instrument, root-canal shaping procedures can be implemented efficiently for patients with limited mouth opening, owing to the bending characteristic of the instrument. The potential risk of abrupt mouth closing due to mouth-opening fatigue or difficulty and the consequent potential risk of instrument fracture are reduced. Additionally, invasive and/or conservative access cavity preparation can be minimized through efficient use of spring-machined instruments. Thus, even in cases where the orifices are far from the center of the tooth, safe root-canal instrumentation can be achieved using spring-machined NiTi rotary instruments.

## 5. Conclusions

Under the conditions of this study, it can be concluded that spring machining on the shank of NiTi rotary instruments provides a stress-bearing area and may attenuate the torsional and cyclic fatigue of the instruments. Spring machining may enhance both the cyclic fatigue resistance and the torsional resistance of NiTi rotary instruments. Additional studies on the effects of spring machining on NiTi rotary instruments with various alloys and geometries, as well as the effects of different types of spring-machining treatments, should be performed in the future.

## Figures and Tables

**Figure 1 materials-13-05246-f001:**
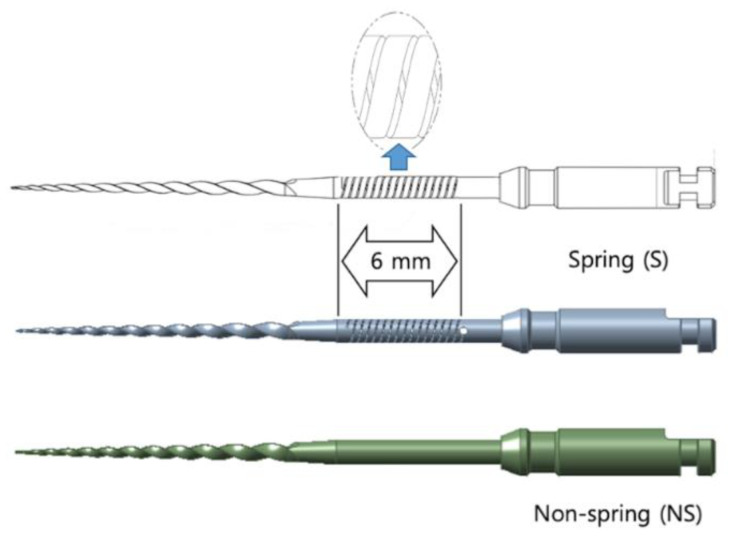
Illustrations of spring file with (S) and without (NS) spring machining.

**Figure 2 materials-13-05246-f002:**
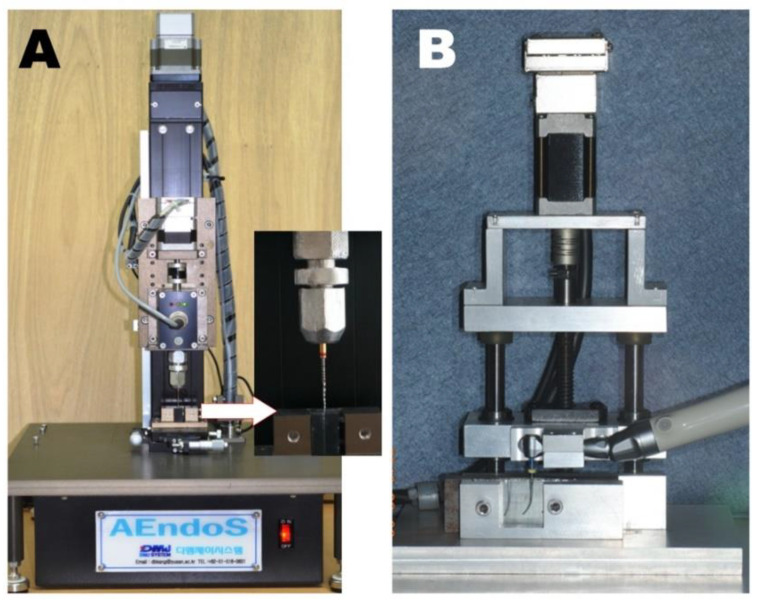
Test devices used in this study. (**A**) AEndoS for the torsional resistance test (the magnified area shows the restriction of the file using polycarbonate resin blocks) and (**B**) EndoC for the cyclic fatigue resistance test.

**Figure 3 materials-13-05246-f003:**
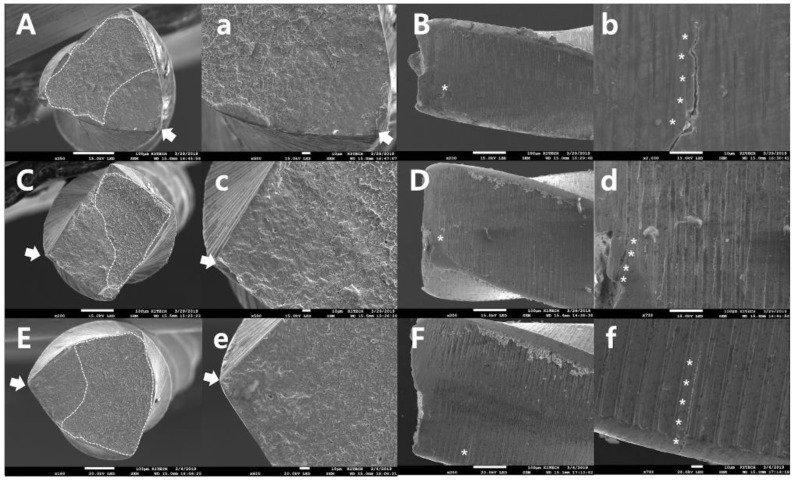
Representative topographic scanning electron microscope images taken after the cyclic fatigue fracture tests (**A**,**B**,**a**,**b** for SPR-S; **C**,**D**,**c**,**d** for PTN-S; **E**,**F**,**e**,**f** for PTG-S; figures labeled with lowercase letters are magnified compared with the figures labeled with uppercase letters). Regardless of the file brands and spring machining, the specimens exhibited a crack-initiation area (arrows) and an overloaded fast fracture zone (dotted area) in the cross-sectional views and had microcracks along the machining groove on the longitudinal aspects.

**Figure 4 materials-13-05246-f004:**
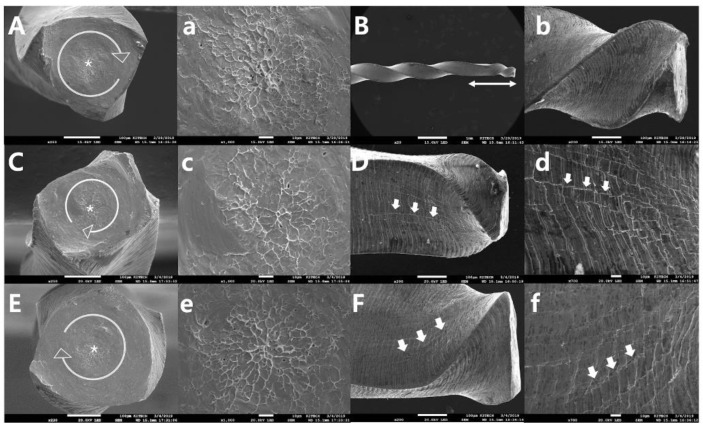
Representative topographic scanning electron microscope images taken after the torsional fracture tests (**A**,**B**,**a**,**b** for SPR-S; **C**,**D**,**c**,**d** for PTN-S; **E**,**F**,**e**,**f** for PTG-S; figures labeled with lowercase letters are magnified compared with the figures labeled with uppercase letters). Regardless of the file brands and spring machining, the specimens exhibited typical features, including circular abrasion marks (circular arrows; **A**–**C**) and skewed dimples (**a**–**c**) near the center of rotation, in the cross-sectional views and had an unwound helix (double ended arrow) of the file flute and slipped microcracks (arrows) at the longitudinal aspects of the file.

**Table 1 materials-13-05246-t001:** Cyclic fatigue resistance of each tested group (mean ± standard deviation).

		Non-Spring (NS)	Spring (S)
NCF *	SPR *	1348 ± 196	1810 ± 410
	PTN *	1618 ± 283	2065 ± 239
	PTG *	2099 ± 147	2623 ± 284
Fracture Fragment Length (mm)	SPR	1.87 ± 0.18	1.84 ± 0.21
	PTN	3.59 ± 0.40	3.48 ± 0.25
	PTG	3.46 ± 0.62	3.20 ± 0.16

SPR: Spring file; PTN: ProTaper Next; PTG: ProTaper Gold. NCF: Number of cycles to failure. * There were significant differences in the number of cycles to failure (NCF) between the non-spring and spring group for each file system (*p* < 0.05). Fracture fragment length did not show any significant difference between non-spring and spring groups (*p* > 0.05).

**Table 2 materials-13-05246-t002:** Torsional resistance of each tested group (mean ± standard deviation).

		Non-Spring (NS)	Spring (S)
Toughness * (°·Ncm)	SPR *	513.82 ± 45.86	565.29 ± 61.80
	PTN *	217.37 ± 26.74	253.26 ± 26.11
	PTG *	544.41 ± 94.72	659.76 ± 100.97
Ultimate Strength (Ncm)	SPR	1.11 ± 0.09	1.11 ± 0.09
	PTN	0.84 ± 0.09	0.82 ± 0.05
	PTG	1.43 ± 0.12	1.48 ± 0.08
Distortion Angle * (°)	SPR *	621.61 ± 28.00	676.23 ± 63.06
	PTN *	359.36 ± 19.51	429.91 ± 26.44
	PTG *	507.76 ± 72.94	641.48 ± 117.64

SPR: Spring file; PTN: ProTaper Next; PTG: ProTaper Gold. * There were significant differences in toughness and distortion angle between the non-spring and spring groups for each file system (*p* < 0.05). No significant difference was shown in ultimate strength between non-spring and spring groups (*p* > 0.05).

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
