# Peer review of "Advancement of Mechanical Properties of Nickel-Titanium Rotary Endodontic Instruments by Spring Machining on the File Shaft"

_materials, 2020, doi:10.3390/ma13225246_

Round 1

Reviewer 1 Report

This original scientific research evaluates the impact from spring machining of the shaft of 3 different files on the cyclic fatigue and torsional resistance of those instruments, compared with regular (non-spring) shafts.

The study is well-designed and addresses an important topic for the field of endodontics, as this can impact on the risk of fracture of the instruments during root canal preparation, and this is a current concern of clinicians. The manuscript is well-written and easy to follow. The methodology is sound and the bibliography well-cited. The findings of this study increase the scientific knowledge in this field.

In this reviewer opinion, introduction needs to be improved, to better present the gap in knowledge intended to be filled by this study. Other efforts have been assessed in order to improve the cyclic fatigue resistance of instruments, as the treatment protocol (Webber et al (2020). Higher Speed and No GlidePath: A New Protocol to increase the efficiency of XP Shaper in curved canals—An in vitro study. JOE) or the surface treatment of the instruments (Palma et al (2019). Cyclic fatigue resistance of three rotary file systems in a dynamic model after immersion in sodium hypochlorite. Odontology). Other factors with impact in the torsional deformation have been studied and must be cited (Isik et al (2020). Effect of Shaft Length on the Torsional Resistance of Rotary Nickel–titanium Instruments. JOE).

At Materials & Methods, please detail the laser cutting parameters used to machine the spring in file shafts.

At discussion, the cyclic fatigue results and the length of fractured fragments observed in the PTN instruments needs to be compare with results from Palma et al 2019. Also discuss the impact of spring on torsional resistance of the file compared to other factors, as observed by Isik et al 2020.

Author Response

This original scientific research evaluates the impact from spring machining of the shaft of 3 different files on the cyclic fatigue and torsional resistance of those instruments, compared with regular (non-spring) shafts.

The study is well-designed and addresses an important topic for the field of endodontics, as this can impact on the risk of fracture of the instruments during root canal preparation, and this is a current concern of clinicians. The manuscript is well-written and easy to follow. The methodology is sound and the bibliography well-cited. The findings of this study increase the scientific knowledge in this field.

Authors’ response: Thank you for this positive review comments.

In this reviewer opinion, introduction needs to be improved, to better present the gap in knowledge intended to be filled by this study. Other efforts have been assessed in order to improve the cyclic fatigue resistance of instruments, as the treatment protocol (Webber et al(2020). Higher Speed and No GlidePath: A New Protocol to increase the efficiency of XP Shaper in curved canals—An in vitro study. JOE) or the surface treatment of the instruments (Palma et al(2019). Cyclic fatigue resistance of three rotary file systems in a dynamic model after immersion in sodium hypochlorite. Odontology). Other factors with impact in the torsional deformation have been studied and must be cited (Isik et al (2020). Effect of Shaft Length on the Torsional Resistance of Rotary Nickel–titanium Instruments. JOE).

Authors’ response: Thank you for the suggestions with references. We have describe further introduction for the idea to increase the mechanical properties with the whole references as suggested.

At Materials & Methods, please detail the laser cutting parameters used to machine the spring in file shafts.

Authors’ response: Thank you. Originally, we wanted to present the technique how to create the spring on the shaft as only with “laser cutting”. But we actually suggested some more detail of the portion; the full cutting rotation angle and the longitudinal length 0f 6 mm. However, it is IMPOSSIBLE to present the parameters used for the machining procedure to protect the patent.

At discussion, the cyclic fatigue results and the length of fractured fragments observed in the PTN instruments needs to be compare with results from Palma et al 2019.

Authors’ response: Thank you for this comment. Their study compared the cyclic fatigue resistance according to the NaOCl immersion in 45 degree curvature, whereas our study used 35 degree. This small difference in curvature angle may make the differences in the cyclic fatigue resistance of the file. While the mean values of NCF were comparable according to the curvature angles (1200 ± 178 vs. 1618 ± 283), the study purpose should be considered. The present study aims to see the effect of spring machining on the fracture resistances. For this, we compared 3 file systems with and without spring machining. Comparing the different parameters does not seem to be appropriate. Thank you.

Also discuss the impact of spring on torsional resistance of the file compared to other factors, as observed by Isik et al 2020.

Authors’ response: Isik et al studied the effect from the length of the shaft and the present study evaluated the effect from spring machining, So that we could not directly compare the results. But we have discussed the effect from shaft portion regardless of the spring existence.

Reviewer 2 Report

This is a scientific work that reports the spring technique on the mechanical and fracture properties of rotating NiTi instruments. Some criticism are present:

- An opening sentence in the abstract section on the problem related to nickel titanium instruments should be inserted

- Keywords are not MESH words in search engines like PUBMED. Authors are asked to modify them all

-The introduction section appears too small and in my opinion it does not completely help the reader to understand the problem. In particular, some properties of NiTi rotary instruments, such as one and two-way elastic memory and superelasticity, should be fully described. was also done on the differences in terms of mechanical properties between these tools and traditional manual ones.

-In the introduction section no precise bibliographical references have been made when mentioning the various attempts made to improve the properties of the instruments

-At the end of the introduction section, the null hypotheses of the study must be inserted and must be refuted at the end of the same.

-No information is provided on the Spring procedure and how it is performed

-What are the reasons for choosing these tools?

What are the changes made to ISO 3620-1? What is the bibliographic support in the choice?

- A paragraph specifically for the SEM section should be chosen to explain how the samples for experimentation were prepared

-A separate paragraph for statistical evaluation should be adjusted

-The captions of Tables 1 and 2 are too long and confusing. Write an asterisk in the table for statistically significant values

-The discussion section does not seem well structured. After the initial sentence, in fact, a concise definition of the results obtained must be absolutely inserted.

- Immediately afterwards, this study must be compared with the numerous other works in the literature which, by modifying the surface of the NITi instruments, have tried to improve their mechanical characteristics. The analysis of the literature in this work appears superficial and incorrect.

-No reference was made, moreover, on the possible methods of defining stress in endodontic instruments. In this regard, I recommend to include in the reference section and in the discussions the following scientific work that could be useful, for example in the citation of ANSYS as a very reliable method of investigation:

-Chieruzzi M, Pagano S, Cianetti S, Lombardo G, Kenny JM, Torre L. Effect of fibre posts, bone losses and fibre content on the biomechanical behaviour of endodontically treated teeth: 3D-finite element analysis. Mater Sci Eng C Mater Biol Appl. 2017 May 1;74:334-346. doi: 10.1016/j.msec.2016.12.022. Epub 2016 Dec 7. PMID: 28254302.

-I invite the authors to explain, also with bibliographical references, the difference in results not only after spring but also between the tools. what characteristics can these results refer to?

- I think the reference section is absolutely not sufficient: too many works are old and a modern and recent bibliographic search does not seem to have been carried out in the realization of the scientific work

Author Response

This is a scientific work that reports the spring technique on the mechanical and fracture properties of rotating NiTi instruments. Some criticism are present:

- An opening sentence in the abstract section on the problem related to nickel titanium instruments should be inserted

Authors’ response: Thank you. We have described short background of the study.

- Keywords are not MESH words in search engines like PUBMED. Authors are asked to modify them all

Authors response: We have replaced the keywords with searched from the MeSH Browser. If editor and editorial office permit to present original words, previous ones would be better to suggest the KEY words. Thank you.

-The introduction section appears too small and in my opinion it does not completely help the reader to understand the problem. In particular, some properties of NiTi rotary instruments, such as one and two-way elastic memory and superelasticity, should be fully described. was also done on the differences in terms of mechanical properties between these tools and traditional manual ones.

Authors response: Thank you for these comments. We have added further detail to present the background of the study. We have submitted this manuscript to the special issue of “Contemporary Endodontic Material” and the basic science presented in the manuscript may enough to bring the clinical concern and related material (instrument / file) for the purpose of the study.

-In the introduction section no precise bibliographical references have been made when mentioning the various attempts made to improve the properties of the instruments

Authors response: Thank you for this notice. We have added the references need for the sentences.

-At the end of the introduction section, the null hypotheses of the study must be inserted and must be refuted at the end of the same.

Authors response: Thank you for the comment. The null hypothesis was added in introduction, and discussion. 

-No information is provided on the Spring procedure and how it is performed

Authors’ response: Thank you. The machining process by using laser was conducted on the shaft part of each NiTi files. Hope to understand it could not be written in fully detail due to the patent protection. The company could only present the laser source and the basic structure as in Figure 1. Thank you.

-What are the reasons for choosing these tools?

What are the changes made to ISO 3620-1? What is the bibliographic support in the choice?

Authors’ response: Thank you for this comment. We have followed the ISO for the test procedure although it is not the manual stainless stain instrument but the rotary instrument rotating in a high speed. The modification is detailed in the following sentences. The polycarbonate resin blocks were used to fix the file tip not to hurt or deform the file structure. The custom device AEndoS is dedicated device for the studies using the instruments under various conditions clinically applicable. Basically, it is very similar with the micro or mini Instron. It can control the rotation speed, angle, pecking speed and depth, rotary or reciprocating movement, and measure the generated torque and vertical pressure, etc. This is the endodontics dedicated device and have lots of publication using it.

- A paragraph specifically for the SEM section should be chosen to explain how the samples for experimentation were prepared

Authors’ response: Thank you for this notice. We have detailed for the section of SEM.

-A separate paragraph for statistical evaluation should be adjusted

Authors’ response: We have structured according to the guideline from editorial. Thank you.

-The captions of Tables 1 and 2 are too long and confusing. Write an asterisk in the table for statistically significant values

Authors’ response: Thank you. We have mended the tables accordingly.

-The discussion section does not seem well structured. After the initial sentence, in fact, a concise definition of the results obtained must be absolutely inserted.

Authors’ response: We have reviewed the discussion section and edited further for better readability. Thank you.

- Immediately afterwards, this study must be compared with the numerous other works in the literature which, by modifying the surface of the NITi instruments, have tried to improve their mechanical characteristics. The analysis of the literature in this work appears superficial and incorrect.

Authors’ response: Unfortunately, this is the first trial to evaluate the effect from spring machining on the shaft and the purpose of this study was to compare the effect from the presence of spring structure. The other attempts and trials to increase the mechanical properties of files were compared in many of previous studies using their appropriate controls, for example, heat treated and not-treated with same geometric structures or among the instruments heat treated. The comparison and following discussion would be proper when the condition and environments are under the controls. Thank you.

-No reference was made, moreover, on the possible methods of defining stress in endodontic instruments. In this regard, I recommend to include in the reference section and in the discussions the following scientific work that could be useful, for example in the citation of ANSYS as a very reliable method of investigation:

-Chieruzzi M, Pagano S, Cianetti S, Lombardo G, Kenny JM, Torre L. Effect of fibre posts, bone losses and fibre content on the biomechanical behaviour of endodontically treated teeth: 3D-finite element analysis. Mater Sci Eng C Mater Biol Appl. 2017 May 1;74:334-346. doi: 10.1016/j.msec.2016.12.022. Epub 2016 Dec 7. PMID: 28254302.

Authors’ response: Thank you for this kind suggestion. We have done this study by using real fracture in vitro test to compare the effect from spring structure. The FEA is proper method to indicate the stress generation points or distribution. We have cited another article in press where the spring structures were verified recently (https://doi.org/10.1016/j.joen.2020.10.015). Thank you.

-I invite the authors to explain, also with bibliographical references, the difference in results not only after spring but also between the tools. what characteristics can these results refer to?

Authors’ response: This study focused on the effectiveness of spring machining. That’s why we compared the parameters before and after spring machining. Other two file system commercially introduced were tested as another verification procedure whether the spring structure really works for other file system. The differences between the file systems are decided by their geometric differences and alloy mainly. We have added these in the discussion section. Thank you.

- I think the reference section is absolutely not sufficient: too many works are old and a modern and recent bibliographic search does not seem to have been carried out in the realization of the scientific work

Authors’ response: Thank you. We agree this notice and updated the references further.

Reviewer 3 Report

Major comments:

1. The overall number of stress cycles performed (15) may not reflect a true resistance/susceptibility to torsional instruments' fracture.

2. The selection of specific type/manufacturer of rotary instruments needs further explanations.

3. The individualized spring adjustment of the shank with laser does not seem to be standardised and comparable.

Minor comments:

1. I would argue that t-students test is optimal for the purpose of statistical analysis.

2. The methodology part requires rephrasing and 'softening' in terms of technical terminology.

Author Response

Major comments:

  1. The overall number of stress cycles performed (15) may not reflect a true resistance/susceptibility to torsional instruments' fracture.

Authors’ response: Thank you for this comment. This standard test method for cyclic fatigue fracrture test usually have the samples between 10 and 15 in numerous published papers. We believe 15 per each file group (total 90) is enough to test. It’s difficult to find an article where used more than 15.

  1. The selection of specific type/manufacturer of rotary instruments needs further explanations.

Authors’ response: Thank you for this comment. We have detailed further for the tested materials, but the two file systems were included to evaluate whether the spring structure really works for other file systems. We are worrying that the too much details for other two commercially marketed may disperse the readers idea. Thank you.

  1. The individualized spring adjustment of the shank with laser does not seem to be standardised and comparable.

Authors’ response: The laser cutting machining is done by the manufacturing company with a dedicated device and controlled of course with computer software. We have described further detail about the spring structure.

Minor comments:

  1. I would argue that t-students test is optimal for the purpose of statistical analysis.

Authors’ response: Thank you for this comment. As noted earlier, the main purpose of this study is to compare the difference between files with and without spring machining NOT among the three file systems. If we add the statistic comparison among three file groups using such a ANOVA and post-hoc may disperse the main purpose.

  1. The methodology part requires rephrasing and 'softening' in terms of technical terminology.

Authors’ response: Thank you for this notice. 

Round 2

Reviewer 2 Report

Revision work is complicated and carefully executed work. The authors who submit themselves to it have the duty to give rational justifications for the requested questions. Replacing a suggestion with a self-quotation, however in press, represents a serious lack of respect for the review itself.

Author Response

Revision work is complicated and carefully executed work. The authors who submit themselves to it have the duty to give rational justifications for the requested questions. Replacing a suggestion with a self-quotation, however in press, represents a serious lack of respect for the review itself.

Authors’ response: We respect the reviewers’ comments and made revision according to the review points. We have added some more references according to the 3 reviewers’ comments and there was one article published from the same groups (not the same authors). We don’t really intended the self-citation but just followed the reviewer #1’ comments. And also for about the references of FEA, we think that the FEA about the spring file itself is much better to understand the contents. This is why we have added the FEA article that is in-press instead of the article about biomechanical behaviour of endodontically treated teeth. We hope kind understanding for this. Thank you.    

Reviewer 3 Report

The robust explanations provided do not preclude the fact that the individualized and non-standardized laser adjustment of the instrument's shank is deemed highly unpredictable and may effect the instrument mechanical properties. This needs to be clearly noted and emphasized in the main text. 

The statistical methods still have to be enhanced/replaced to better portrait obtained results. 

Author Response

The robust explanations provided do not preclude the fact that the individualized and non-standardized laser adjustment of the instrument's shank is deemed highly unpredictable and may effect the instrument mechanical properties. This needs to be clearly noted and emphasized in the main text. 

Authors’ response: We have replied on the comment of 1st review process as the laser cutting machining is done by the manufacturing company (Denflex, Seoul, Korea) with a dedicated device and controlled of course with computer software. This method absolutely guarantees the standardizing machining process including coil depth and angle of coil rotation, etc. So that the manufacturer can produce the brand file. We may not say unpredictable for the commercially available product. Thank you.  

The statistical methods still have to be enhanced/replaced to better portrait obtained results. 

Authors’ response: Thank you. We recognize that there are two variables in the test; the presence of spring and file brand. In this study, authors dis not intended to compare the resistance among the files, but tried to see the effect from the spring in the same file brand. PTG and PTN were tested to see the effect from the spring when it was applied to the brand files.

This manuscript is a resubmission of an earlier submission. The following is a list of the peer review reports and author responses from that submission.

Round 1

Reviewer 1 Report

This original scientific research evaluates the impact from spring machining of the shaft of 3 different files on the cyclic fatigue and torsional resistance of those instruments, compared with regular (non-spring) shafts.

The study is well-designed and addresses an important topic for the field of endodontics, as this can impact on the risk of fracture of the instruments during root canal preparation, and this is a current concern of clinicians. The manuscript is well-written and easy to follow. The methodology is sound and the bibliography well-cited. The findings of this study increase the scientific knowledge in this field.

In this reviewer opinion, introduction is the only section which needs to be improved, to better present the gap in knowledge intended to be filled by this study. Other efforts have been assessed in order to improve the cyclic fatigue resistance of instruments, as the treatment protocol (Webber M. et al (2020). Higher Speed and No GlidePath: A New Protocol to increase the efficiency of XP Shaper in curved canals—An in vitro study. Journal of Endodontics) or the surface treatment of the instruments (Palma P. et al (2019). Cyclic fatigue resistance of three rotary file systems in a dynamic model after immersion in sodium hypochlorite. Odontology). One or two more recent studies also need to be presented about torsional resistance. Although the bibliography presented is well-cited, it should be improved with some more recent publications.

Reviewer 2 Report

The paper describes evaluation of the fatigue and torsional resistance of nickel titanium rotary instruments. The spring machining performed on the shank of the tools increases the compliance of the tool and therefore also its toughness. The instrument strength is unaffected as it is solely based on the file geometry and material. The spring machined part decreases the magnitude the dynamic stress peaks introduced by the function of the instrument tip and therefore also the fatigue resistance of the tool is increased. Note that the above properties (toughness, strength and fatigue resistance) are characterizing the instrument and don’t represent standard material properties as used in material science.

The paper does not characterize the material, but the final product. The performed test are not directly related to the used material which is not specified anyway. Loading during fatigue test is not clear, the channel material is not specified, neither is the loading imposed on the tool tip. The paper is therefore relevant to the dental community but does not provide interesting information to material science community.

Therefore the reviewer recommend to Reject the paper and suggest alternate journal that is more focused on mechanics (while addressing the above points) or dentistry.

Reviewer 3 Report

The submitted manuscript entitled ‘Effects of spring machining on mechanical properties of nickel-titanium rotary instruments’ is dealing with the investigation of three different drillers used in dentistry. The manuscript is interesting, but in the opinion of this Reviewer is rather out-of-scope for the journal Materials. The main aim was to emphasize the beneficial effects of spring machining on the mechanical properties of the drillers, but the results are not convincing, and more measurements may be needed. Another concern is in the lack of exact description of laser machining, surface roughness and other parameters. The setups of the applied non-standardised mechanical tests are also missing.

- Please indicate the main differences between the three different investigated equipment.

- Label the subfigs of fig 1 as (a), (b), etc.

- Please detail the laser cutting parameters used to machine the drillers.

- ‘The ultimate strength (Ncm)…’ – Ncm is not a strength value (MPa), please reconsider. Please use SI units: Nm.

- Toughness can be interpreted in the energy dimension, but definitely not in Ncm, rather in J.

- Please provide sketches about the testing methods since they are not standardized.

- English is not the native language of this Reviewer, but a proof-reading is strongly recommended.